# Investigations of Fish Assemblages Using Two Methods in Three Terminal Reservoirs of the East Route of South-to-North Water Transfer Project, China

**DOI:** 10.3390/ani13101614

**Published:** 2023-05-11

**Authors:** Huiguo Yan, Sibao Chen, Xia Liu, Zhenhao Cheng, Bjorn Victor Schmidt, Wenping He, Fei Cheng, Songguang Xie

**Affiliations:** 1Key Laboratory of Freshwater Fish Reproduction and Development (Ministry of Education), College of Fisheries, Southwest University, Chongqing 400715, China; yanhuiguo1999@163.com; 2Key Laboratory of Aquatic Biodiversity and Conservation of Chinese Academy of Sciences, Institute of Hydrobiology, Chinese Academy of Sciences, Wuhan 430071, China; 3Changjiang Institute of Survey Planning Design and Research, Key Laboratory of Changjiang Regulation and Protection of Ministry of Water Resources, Wuhan 430010, China; 4Shandong Main Line Co., Ltd. of East Route of South-to-North Water Transfer Project, Jinan 250013, China; 5Department of Biological and Environmental Sciences, Texas A&M University, Commerce, TX 77843, USA; 6State Key Laboratory of Marine Resources Utilization in South China Sea, Hainan University, Haikou 570228, China; xiesg@hainanu.edu.cn

**Keywords:** eDNA metabarcoding, fish landing, fish invasion, terminal reservoir, water quality management, water transfer project

## Abstract

**Simple Summary:**

The water quality of reservoirs in water transfer projects is key as these reservoirs provide water directly to people. In general, the composition and structure of fish assemblage can reflect reservoir water quality and can also be regulated for its improving. We compared monitoring results of fish assemblages carried out by the traditional and environmental DNA methods in three-terminal reservoirs of the South-to-North Water Transfer project in China. The fish assemblages determined by both methods showed similar assemblage structure and patterns of diversity and spatial distribution; however, obvious differences in fish composition across the three reservoirs examined were found. Demersal and small fishes were the most common in all sites. Moreover, a strong association between water transfer distance and the distribution of non-native fish was found. Our findings highlight the importance of fish assemblage monitoring and the impact of water transfer distance on fish assemblage, providing valuable information that can be used in future water transfer projects.

**Abstract:**

The terminal reservoirs of water transfer projects directly supply water for domestic, agricultural, and industrial applications, and the water quality of these reservoirs produce crucial effects on the achievement of project targets. Typically, fish assemblages are monitored as indicators of reservoir water quality, and can also be regulated for its improvement. In the present study, we compared traditional fish landing (TFL) and environmental DNA (eDNA) metabarcoding methods for monitoring fish assemblages in three terminal reservoirs of the East Route of the South-to-North Water Transfer Project, China. Results of TFL and eDNA showed similar assemblage structures and patterns of diversity and spatial distribution with obvious differences in fish composition across three examined reservoirs. Demersal and small fish were dominant in all reservoirs. In addition, a strong association between water transfer distance and assemblages and distribution of non-native fish was found. Our findings highlight the necessity of the fish assemblage monitoring and managing for water quality and revealed the impact of water diversion distance on the structure of fish assemblages and dispersal of alien species along the water transfer project.

## 1. Introduction

Water transfer projects are one of the major pathways for resolving the shortage stress of water resources, and they have been widely implemented [1,2,3]. It is considered that approximately 25% of the world’s freshwater resources are diverted by over 170 inter-basin water transfer projects that presently exist [4]. The South-to-North Water Transfer Project (SNWTP) of China—the largest inter-basin water transfer project with an annual capacity of 1827 billion m^3^—directly benefits 110 million people living in areas with severe water scarcity [5]. Typically, water transfer projects have a series of terminal reservoirs that directly store and supply diverted water for domestic, agricultural, and industrial applications in recipient regions [6]. Therefore, the water quality management of terminal reservoirs is crucial for achieving the goals of water transfer projects.

The eastern route of SNWTP transfers water with an annual capacity of 8.9 billion m^3^, and its diversion path spans over 1300 km [7]. Typical plain lake terminal reservoirs are important components of the eastern route. Most reservoirs were newly built for the project and present homogeneous physical structures, such as a regular shape, similar depth, and comparable storage capacity, as well as a unified structure design [8]. Except for rainfall, diverted water is the only source of these reservoirs [9]. Therefore, the ecosystem of terminal reservoirs is seriously affected by artificial management and is relatively independent of the surrounding aquatic ecosystems.

Owing to water quality transformation combined with long-term storage and conversion of pollutants along the transfer route, the diverted water may pose a high risk to the water quality of final storage reservoirs [10]. Variations in the species composition, richness, and the structure of fish assemblages may serve as indicators of aquatic ecosystem health and may be associated with the physicochemical conditions of water quality in terminal reservoirs [11,12]. In most terminal reservoirs, fish assemblages are periodically monitored as part of general management. Furthermore, the prevention and control of algal blooms is a major concern in the management of terminal reservoirs, particularly those with drinking water functions. The abundance of planktivorous fish, such as *Hypophthalmichthys molitrix* and *Aristichthys nobilis*, can be regulated as a biomanipulation to control bloom risk in terminal reservoirs [13]. Population parameters of planktivorous fishes, such as age, growth, and feeding, which are important to the biomanipulation, must be assessed through fish monitoring of final storage reservoirs [14]. It is valuable to determine appropriate monitoring methods of fish assemblage in these reservoirs [15].

Traditional methods of fish assemblage monitoring rely on capture-based sampling techniques through netting, electrofishing, and trapping [15,16,17]. While these methods may yield unequivocal information on the occurrence of a species, they often fall short of capturing its full richness and diversity [18,19]. Capture-based methods have additional limitations, including their high cost, labor-intensive nature, and destructive impact on studied populations. Furthermore, successful implementation of these methods typically requires a high degree of taxonomic expertise [20,21]. Thanks to advances in environmental DNA (eDNA) metabarcoding, batch detection of species composition and relative abundance is possible, paving the way for the characterization of fish assemblages in aquatic ecosystems [22]. eDNA metabarcoding is a noninvasive, efficient, and cost-effective biomonitoring method that has been successfully applied for biodiversity assessment of different aquatic ecosystems [23,24,25]. Furthermore, this approach holds great potential for assessing fish abundance and biomass, indicating that it is to be comparable or even superior to conventional capture-based techniques for monitoring fish assemblages in final storage reservoirs [26]. However, eDNA metabarcoding also has some disadvantages such as PCR inhibition, dependent on reference database, and false positives [27].

To this end, in the present study, we compared the species composition, diversity, and assemblage structure of fish detected using traditional fish landing (TFL) and eDNA metabarcoding methods in three terminal reservoirs of the eastern route of the SNWTP. The aims were (1) to validate the congruence between TFL and eDNA metabarcoding methods and verify their applicability for fish assemblage monitoring in terminal reservoirs which can be used to support the management of terminal reservoirs along water transfer projects, and (2) to provide fundamental information on fish assemblages of the three reservoirs that can be used in future actions aimed at water quality improvement of the eastern route.

## 2. Materials and Methods

### 2.1. Study Area

This study was carried out at three reservoirs located along the eastern route of the South-to-North Water Transfer Project (SNWTP), namely (1) Datun (DT), (2) Donghu (DH), and (3) Shuangwangcheng (SWC). These reservoirs were commissioned in 2013 and primarily receive water diverted from Dongping Lake. These storage reservoirs are similar in terms of acreage, operational storage capacity, and structural design (Figure 1). Specifically, the acreages and operational storage capacities of DT, DH, and SWC are, respectively, 5.1, 4.4, and 6.4 km^2^ and, respectively, 4.46, 4.70, and 5.32 billion m^3^. Every reservoir has one inlet for receiving the diverted water and two outlets for water supply to the recipient regions. DH and SWC share an identical pathway of water diversion, with a straight distance of approximately 130 and 257 km from Dongping Lake, respectively. DT is located on another pathway, with a straight distance of approximately 130 km from the lake (Figure 1).

Three sampling sites were set up for both TFL and eDNA surveys in each reservoir, and the surveys were conducted in October 2021 (Figure 1).

### 2.2. Samplings Processing of TFL

To effectively collect fish of various sizes that inhabit different depths, TFL sampling was applied using a combination net of multi-mesh gillnet and trap-net. The multi-mesh gillnet comprised 12 panels of different mesh sizes, i.e., ranging from 5 to 55 mm. Each gillnet was 30 m long and 1.5 m high, with 2.5 m-long mesh panels. Mesh panels were assembled randomly and maintained in the same order in all multi-mesh gillnets [28]. The trap-net comprised a single mesh size (4 mm) and was 18 m long, 0.45 m wide, and 0.33 m high. The trap net was used without baits.

Three combination nets (one multi-mesh gillnet and one trap-net) were installed simultaneously at the three sampling sites in each reservoir. The combination net was placed in the littoral zone and exposed for 12 h from dusk to the next morning [29]. The multi-mesh gillnets were deployed at a water depth of approximately 7 m in a straight line roughly parallel to the shoreline, and the trap nets were placed perpendicular to the shoreline.

After the catch, all fish were retrieved from the nets and then identified to the species according to their morphological characteristics [30,31,32,33]. All fish were sorted and counted by species, and the standard body length (precision = 0.1 cm) and body weight (precision = 0.01 g) of each individual were determined.

### 2.3. eDNA Sample Processing

For eDNA analysis, water samples were simultaneously collected from the three sites before placing the nets. From each site, 2 L of water was collected from the pelagic and bottom layers (half each) at the gillnet location using sterile vials (Thermo Fisher Scientific™, Waltham, MA, USA). Blank samples were established using 2 L of distilled and deionized water to check for contamination during field collection. After collection, all samples were immediately chilled with ice and kept at a low temperature until they were delivered to the laboratory.

Each water sample was filtered through a 0.45 μm MCE membrane with 47 mm diameter (Whatman, England) using a vacuum pump. Filtration was completed within 24 h after the collection; the filters were packed into 5 mL sterile centrifuge tubes and immediately stored at −20 °C until DNA extraction. The equipment was rinsed with water and sterilized by soaking in 10% bleach solution for 10 min before each filtering.

### 2.4. Laboratory Processing of eDNA Samples

The PowerWater DNA Isolation Kit (Mobio, AL, USA) was used to extract DNA from the filter membranes following the manufacturer’s instructions. A clean membrane was used as the negative control [34].

A hypervariable fragment of the mitochondrial 12S rRNA gene was PCR amplified using the MiFishU primer [35]. Amplifications were performed in triplicates using 20 μL PCR mixture consisting of 1× FastPfu Buffer, 0.25 mM dNTPs, 0.2 μM forward and reverse primers, 1 U of FastPfu Polymerase (TransGen Biotech Co., Beijing, China), and 10 ng of template DNA. All negative controls were also subjected to PCR amplification along with the samples.

The amplification products were separated by electrophoresis on a 2% agarose gel, extracted, and purified using the AxyPrep DNA Gel Extraction Kit (Axygen Biosciences, Union City, CA, USA) according to the manufacturer’s instructions. The purified PCR products were then quantified using the PicoGreen dsDNA Assay Kit (Invitrogen, Waltham, MA, USA). Following quantification, the amplicons were pooled at equal amounts, and paired-end 2 × 250 bp sequencing was performed using the Illumina MiSeq platform at Shanghai BIOZERON Biotechnology Ltd. (Shanghai, China) according to standard protocols [36].

Sequencing data were processed using the Quantitative Insights into Microbial Ecology (QIIME, v1.8.0) pipeline [37]. Raw sequence data in FASTQ format file from high-throughput sequencing were initially screened, and low-quality sequences were filtered [36]. Chimeras of the raw sequence data were identified using Perseus [38]. Singletons, the sequences that occur only once in the whole dataset, usually represent PCR or sequencing artifacts [39,40]. Singletons, sequences with ambiguous bases, and the identified chimeras were removed using Mothur 1.36.1 [41]. The sequence outputs from Mothur software were used to assemble paired-end reads using FLASH [42]. Furthermore, the assembled reads that exactly matched the amplified target sequences were determined to be valid sequences using the QIIME pipeline and then assigned to their respective samples [36].

Finally, high-quality sequences were clustered at 97% sequence identity to generate the representative sequences of operational taxonomic units (OTUs) using UCLUST [43]. The OTUs were compared with sequences from GenBank, and OTUs sharing ≥97% identity and an E-value threshold of 10^−5^ among the top BLAST hit sequences were assigned to the species level [35]. Each identified species was confirmed based on information about its distribution available in FishBase (http://www.fishbase.org/, 25 October 2021). After data processing, lists of fish species were determined for each sample site.

### 2.5. Data Analysis

To confirm the consistency between eDNA and TFL, the species composition and dominance were estimated for each method and then compared to each other. For this purpose, all fish were classified based on their feeding habits and inhabited water depth. Feeding habits included planktivorous, herbivorous, omnivorous, or carnivorous species, and inhabiting water depths included the upper, lower, or demersal water layers [44]. The small body size of fish was determined following the criteria described by Lyons [45].

The dominant species in the TFL collection were determined according to the relative importance index (IRI), calculated for each species using the following equation:IRI = (N% + W%) × F%,
where N% and W% are the relative abundance and weight percentages of the given species in the total catch, respectively, and F% represents the occurrence percentage of each species in a given reservoir. When the IRI of a given species exceeds 1000, the species is defined as dominant [46].

Dominant species based on eDNA detection were determined through relative abundance by dividing the read number of a given species by the total number of reads for all species [47]. Species with relative abundance >5% were defined as dominant [48].

Furthermore, four diversity indices were calculated for each reservoir using TFL and eDNA detection data separately. Specifically, the Shannon–Wiener index [49], Pielou’s evenness index [50], Chao1 index [51] and ACE index [52] were calculated following the previously established equations [53].

To visualize the differences in fish assemblages detected by the two methods, a Venn plot was constructed to indicate the overlap of the species number between TFL and eDNA metabarcoding [54], and heat maps were plotted to present the variations in the dominant species of the fish assemblage based on the relative abundance of each species [55]. Non-metric multidimensional scaling (nMDS) was applied to visually assess the similarity of fish assemblages among the three reservoirs based on Bray–Curtis similarity matrices [56]. The reliability of nMDS analysis was evaluated using stress coefficients. High precision without the real risk of drawing wrong inferences is indicated when the coefficient is <0.1 [57].

The sampling map of the study area was drawn using ArcGIS 10.8. The diversity indices, Venn plot, and heat maps were determined using vegan, VennDiagram, and pheatmap packages in R 4.1.3, respectively [58]. The nMDS analysis was performed using the metaMDS function of the vegan package in R 4.1.3 [59].

## 3. Results

### 3.1. Fish Assemblages Identified through Traditional Fish Landing (TFL) Analysis

A total of 1093 fish were collected from three examined reservoirs, which were classified into nineteen species, belonging to four orders, six families, and nineteen genera (Table 1). Cyprinidae accounted for the highest number of fish species with thirteen species. Among the species identified, five were planktivorous, seven were omnivorous, and seven were carnivorous. Additionally, seven species were found in the upper water layer, one in the lower layer, and eleven in the demersal layer. In addition, out of the total identified species, fourteen (73.68%) were categorized as small-sized (Table 1).

In terms of each reservoir, 14, 14, and 11 species were collected from DT, DH, and SWC, respectively (Table 1). Planktivorous fish were the least frequent in all reservoirs; omnivorous fish were the most abundant in DT (50.00% of species) and DH (42.86% of species), and carnivorous fish were the most frequent in SWC (54.55% of species). The majority of fish within the assemblages were small and demersal species. As such, demersal fish accounted for 71.43%, 64.29%, and 72.73%, while small fish made up 92.86%, 71.43%, and 72.73% of the total fish number in DT, DH, and SWC, respectively (Table 1). The IRI defined six, five, and four species as dominant in DT, DH, and SWC, respectively (Table 2). Most of the dominant species, such as *Tridentiger bifasciatus*, *Abbottina rivularis*, and *Pseudorasbora parva*, were small fish, accounting for 88.70%, 84.26%, and 65.00% of the total number of fish collected from DT, DH, and SWC (Table 2).

### 3.2. Fish Assemblages Detected through eDNA Metabarcoding

In total, 5,192,292 sequence reads were obtained and 1,251,754 OTUs were detected. Only 433,521 OTUs (34.63% of the total) were assigned to 20 taxonomic species. These 20 species belonged to 5 orders, 6 families, and 18 genera (Table 1). Cyprinidae accounted for the highest number of fish. The identified species were categorized into four feeding groups, with carnivorous fish being the most common, comprising eight species, while herbivorous fish were the least common, consisting of only two species. Additionally, seven, three, and ten species were found to inhabit the upper, lower, and demersal water layers, respectively. Furthermore, thirteen species were classified as small fish (Table 1).

When considering each reservoir individually, 18, 18, and 17 species were detected in DT, DH, and SWC, respectively (Table 1). Herbivorous and planktivorous fish were less frequent in all reservoirs. For instance, only one herbivorous and four planktivorous species were detected in DT and DH, respectively. Carnivorous species were the most frequent, accounting for 44.44%, 38.89%, and 41.18% of the total species in DT, DH, and SWC, respectively. Demersal and small fish were the major components of assemblage structures in all reservoirs. Demersal fish accounted for 50.00%, 55.56%, and 52.94% and small fish for 66.67%, 72.22%, and 70.59% of the total number in DT, DH, and SWC, respectively (Table 1). Based on the relative abundance, five, five, and three dominant species were identified in DT, DH, and SWC (Figure 2). Most of the dominant species, such as *T*. *bifasciatus*, *Hemiculter leucisculus*, and *Rhinogobius giurinus*, were small fish, accounting for 85.30%, 53.52%, and 94.97% of the total number of identified fish in DT, DH, and SWC.

### 3.3. Comparison of Traditional Fish Landing (TFL) and eDNA Metabarcoding Methods

In total, 26 species were detected across the three reservoirs, with 13 species identified using both methods simultaneously (Table 1). Six species were detected only with TFL, including two species inhabiting the upper water layers and four demersal species. Meanwhile, seven species were only detected with eDNA metabarcoding, including two herbivorous species, two species inhabiting the upper water layers, and three demersal species (Table 1).

In individual reservoirs, 24, 22 and 23 species were collectively detected using the two methods in DT, DH and SWC, respectively (Figure 3). The concordance of species detected simultaneously of eDNA with TFL varied between 21.7 and 45.5% in the three reservoirs. In general, the number of species uniquely detected by eDNA metabarcoding was higher than that uniquely detected by TFL. For instance, 41.70%, 36.40%, and 52.20% of the species were only detected by eDNA metabarcoding in DT, DH, and SWC (Figure 3). The dominant species detected by either method were similar among the reservoirs, except for SWC (Figure 4). In DT and DH, small fish such as *H. leucisculus*, *T. bifasciatus*, and *Acheilognathus macropterus* were the dominant species (Figure 4).

Overall, the two methods revealed a consistent variability pattern in the three reservoirs, and the calculated diversity indices were the highest in DT and the lowest in SWC (Table 3). The Shannon–Wiener and Pielou’s evenness indices were higher for TFL, whereas Chao1 and ACE indices were higher for eDNA metabarcoding (Table 3). nMDS analysis showed a similar spatial pattern of fish assemblages for both methods across the examined reservoirs, with the highest similarity between DT and DH (Figure 5).

## 4. Discussion

From our results, it can be seen that traditional fish landing (TFL) can be used to monitor fish assemblages in terminal reservoirs, and it is an indispensable method when the management of terminal reservoirs focuses on biomanipulation aimed for prevention and control of algal blooms. TFL analysis revealed clear characteristics and patterns in the fish assemblage structures among the three reservoirs, which supports its use as a method for routinely monitoring fish populations in terminal reservoirs. Furthermore, the TFL method collects biological data, such as body length and body weight, and identifies material for age and feeding. This additional information allows further biological studies and provides fundamental data for biomanipulation [60]. Meanwhile, prevention and control of algal blooms through biomanipulation is a major concern in the terminal reservoir management of water transfer projects [61]. Therefore, TFL is indispensable for monitoring fish assemblages if the terminal reservoirs require biomanipulation for water quality improvement actions. However, the disadvantages of TFL are obvious, such as high labor intensity, destructive nature, and dependence on taxonomic specialists [47]. For example, in the present study, *Acheilognathus chankaensis* and *Rhinogobius cliffordpopei* were detected by eDNA metabarcoding and were not identified by TFL. Previous investigations recorded and confirmed these two fishes in lakes and reservoirs of the SNWTP [28,62]. The results are likely to be an inaccurate classification because of difficulties in the morphological distinction between *A. macropterus* and *A. chankaensis* or between *R. giurinus* and *R. cliffordpopei*. Such taxonomic difficulties, combined with other disadvantages of TFL, render the general management of terminal reservoirs for fish monitoring highly challenging.

On the other hand, eDNA metabarcoding seems to be suitable for routine fish monitoring for knowing general diversity performance in terminal reservoirs of water transfer projects. Notably, eDNA metabarcoding revealed a higher species number and more ecological types of fish (e.g., herbivorous species). While these artificial reservoirs were determined to lack aquatic vascular plants, our subsequent TFL identified herbivorous fish, including *Ctenopharyngodon idella* and *Megalobrama amblycephala*. These species may occur because of water diversion and fish stocking. These results demonstrate the feasibility of using eDNA metabarcoding for monitoring fish diversity [63,64]. The performance of eDNA metabarcoding in monitoring fish diversity was tested and supported with essentially similar and moderate levels of congruence to that of the traditional survey method by many references [26,65,66]. Combining other advantages, eDNA metabarcoding is a suitable method for general fish monitoring in terminal reservoirs. Our results indicated obviously different species compositions of fishes between the two methods, which may suggest certain limitations to the application of eDNA metabarcoding fish assemblage monitoring in the terminal reservoirs. The low congruence in species composition between eDNA metabarcoding and TFL results may be related to deficiencies in the taxonomic database used for eDNA referencing. Sequences in public databases are incomplete and seriously biased towards specific geographical areas [67,68], which highlights the importance of developing a local database to improve the applicability of eDNA metabarcoding for fish assemblage monitoring in terminal reservoirs.

In the present study, TFL and eDNA metabarcoding revealed obviously different species compositions of fishes, and more species could be detected if the two methods are used in combination. Therefore, a combined approach including TFL and eDNA metabarcoding may provide more comprehensive information on fish assemblages in terminal reservoirs. Previous studies have reported differences in species detected using conventional methods and eDNA surveys, and found the difficulty in better understanding of how different methods may conflict with each other [47,54,69]. The quantitative metrics in the two-method comparison are urgently needed, which would more straightforwardly reveal differences between them.

Furthermore, our findings highlight the necessity of managing fish assemblage to improve water quality in the three reservoirs. Both methods showed that the fish assemblages of the reservoirs are dominated by demersal and small fish. Specifically, demersal and small fish accounted for over 70% of all fish, whereas planktivorous fish (*H. molitrix* and *A. nobilis*) were rare in all reservoirs. This is worrisome in the reservoirs that managed to reduce algal bloom risk, as demersal fish may intensively destabilize sediment and increase nutrient supply for algal growth [70]. Meanwhile, small fish inhabiting the upper water layers typically feed on zooplankton, thereby decreasing zooplankton control strength on algal density [71]. Finally, in particular, the scarcity of *H. molitrix* and *A. nobilis* in the assemblage may impede fish-based algal control [13]. In addition, our results revealed a relatively simple structure and low diversity of fish assemblages in the reservoirs. The reservoirs under study are newly created artificial systems that operate relatively independently from the surrounding aquatic ecosystems [72]. This makes them particularly conducive to effective fish assemblage management, as the conditions within these reservoirs can be more easily controlled [73]. Based on our results, to ensure water quality safety in the three reservoirs, fish assemblages must be regulated by releasing piscivorous species to control small fish densities, harvesting demersal fish, and increasing *H. molitrix* and *A. nobilis* abundances. Interestingly, our results revealed a clear spatial pattern of fish assemblages among the three reservoirs, which likely reflects the effect of water diversion distance on the structure of fish assemblages in terminal reservoirs. Compared with similar fish assemblages in DT and DH, those in SWC showed distinct dominant species, poor diversity, and different structures. Fish introduction accompanying the water diverted from Dongping Lake is the primary source of fish in the three reservoirs [28]. As opposed to similar water diversion distances between DT and DH, SWC shares an identical diversion pathway but double the linear water diversion distance from the lake of that of DH. Furthermore, the non-native species *T. bifasciatus* was dominant in fish assemblages of DT and DH and rare in fish assemblages of SWC according to both methods. Therefore, water diversion distance may produce crucial effects on the colonization and dispersal of alien species in water bodies under water transfer projects. On the eastern route of SNWTP, the dispersal of two gobiid species, namely *T. bifasciatus* and *Taenioides cirratus*, from the Yangtze Estuary has been reported in several lakes along the route, and the impacts of these carnivorous species warrant special attention [74]. Our results indicate that the management of fish assemblages in terminal reservoirs should consider spatial factors, and monitoring of these reservoirs can offer an excellent opportunity to explore the mechanisms of fish assembly and invasion in water transfer projects.

## 5. Conclusions

In summary, although further research is warranted, our primary results demonstrate the applicability of TFL and eDNA metabarcoding to monitor fish assemblages in terminal reservoirs. Our findings indicate that the fish assemblages identified by both methods exhibited comparable assemblage structures, diversity patterns, and spatial distributions. However, there were noticeable differences in the fish composition observed across the three reservoirs under examination. A combined approach that includes TFL and eDNA metabarcoding may provide more comprehensive information on fish assemblages in terminal reservoirs. Information obtained by both methods underscores the necessity of optimizing fish assemblages to ensure the quality and safety of water in the terminal reservoirs of SNWTP and further hints at the impact of water diversion distance on fish assembly and alien species dispersal along the transfer route. Managing the water quality of terminal reservoirs is crucial for achieving the goals of water transfer projects. Our findings highlight the importance of monitoring fish assemblage in terminal reservoirs and the impact of water diversion distance on the dispersal abilities of non-native fish species, providing valuable information for other water transfer projects.

## Figures and Tables

**Figure 1 animals-13-01614-f001:**
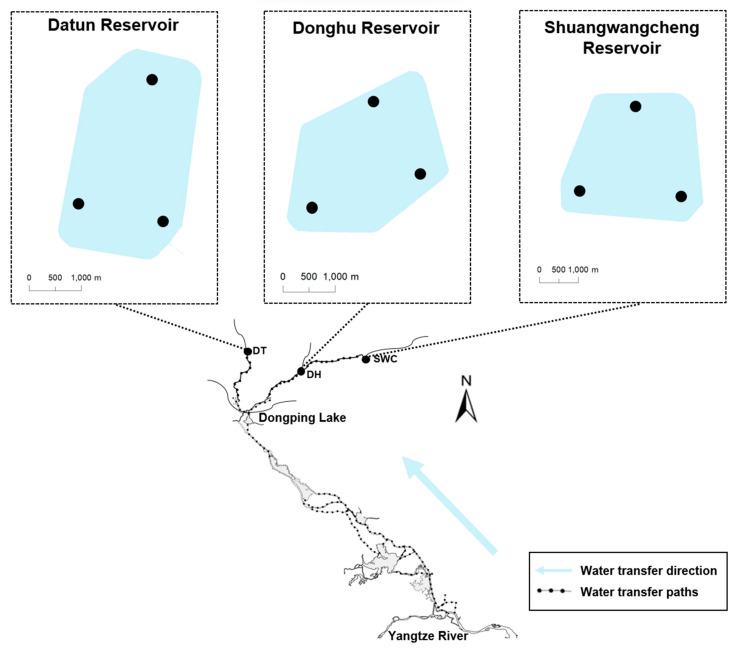
Sampling map of the study area. Solid circles indicate sampling sites.

**Figure 2 animals-13-01614-f002:**
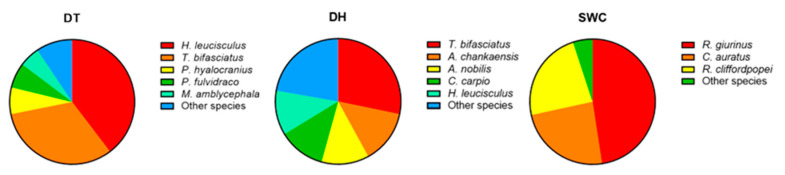
Dominant species detected by environmental DNA (eDNA) metabarcoding in Datun (DT), Donghu (DH), and Shuangwangcheng (SWC) reservoirs.

**Figure 3 animals-13-01614-f003:**
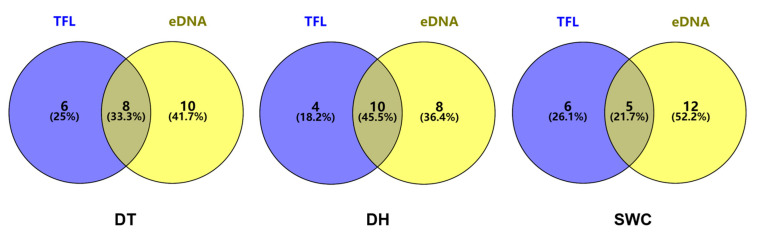
The concordance between traditional fish landing (TFL) and environmental DNA (eDNA) metabarcoding in detecting species across the examined reservoirs.

**Figure 4 animals-13-01614-f004:**
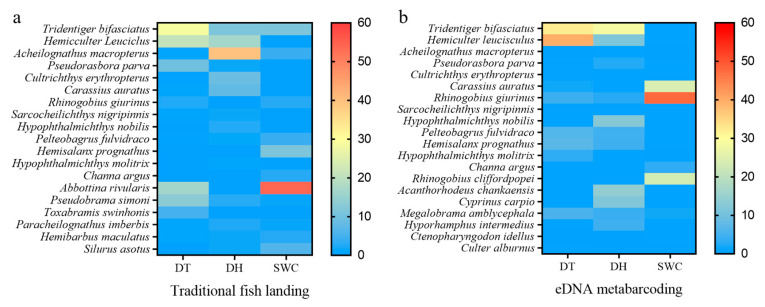
Heat maps of the relative abundance of each species in Datun (DT), Donghu (DH), and Shuangwangcheng (SWC) reservoirs detected by traditional fish landing (TFL, (**a**)) and environmental DNA (eDNA, (**b**)) metabarcoding.

**Figure 5 animals-13-01614-f005:**
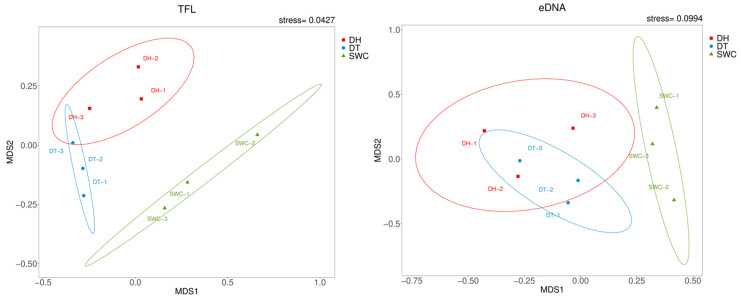
Nonmetric multidimensional scaling (nMDS) analysis of traditional fish landing (TFL) and environmental DNA (eDNA) metabarcoding. In nMDS analysis, the ellipse shows the 95% confidence interval for sampling sites calculated for each reservoir.

**Table 1 animals-13-01614-t001:** Relative abundance (N%), feeding habits, and inhabiting water layers of fish detected by traditional fish landing (TFL) and environmental DNA (eDNA) metabarcoding in three terminal reservoirs of the eastern route of the South-to-North Water Transfer Project.

Orders	Families	Species	Feeding Habits	Water Layer	The Relative Abundance (N%) of TFL			The Relative Abundance (N%) of eDNA		
					DT	DH	SWC	DT	DH	SWC
Salmoniformes	Salangidae	*Protosalanx hyalocranius **	C	UL	0.00	0.00	11.25	6.97	4.29	0.01
Cypriniformes	Cyprinidae	*Ctenopharyngodon idella*	H	LL	0.00	0.00	0.00	0.00	0.00	0.03
		*Toxabramis swinhonis **	P	UL	4.85	0.00	0.00	0.00	0.00	0.00
		*Hemiculter leucisculus **	P	UL	20.84	16.75	0.00	39.66	11.50	0.07
		*Cultrichthys erythropterus **	C	UL	0.16	9.14	0.00	<0.01	<0.01	<0.01
		*Culter alburnus*	C	UL	0.00	0.00	0.00	0.01	0.00	0.00
		*Megalobrama amblycephala*	H	LL	0.00	0.00	0.00	5.33	3.79	1.68
		*Pseudobrama simoni **	P	UL	13.09	2.79	1.25	0.00	0.00	0.00
		*Acheilognathus macropterus **	O	DM	0.81	39.59	3.75	0.00	<0.01	0.00
		*Acheilognathus chankaensis **	O	DM	0.00	0.00	0.00	0.01	13.77	<0.01
		*Paracheilognathus imberbis **	O	DM	0.81	2.03	1.25	0.00	0.00	0.00
		*Hemibarbus maculatus*	O	DM	0.65	1.52	2.50	0.00	0.00	0.00
		*Pseudorasbora parva **	O	DM	9.53	1.52	0.00	0.50	2.61	0.02
		*Sarcocheilichthys nigripinnis **	O	DM	0.81	1.52	0.00	<0.01	0.04	0.01
		*Abbottina rivularis **	O	DM	16.48	0.00	53.75	0.00	0.00	0.00
		*Cyprinus carpio*	O	DM	0.00	0.00	0.00	0.01	11.97	0.01
		*Carassius auratus **	O	DM	0.81	7.61	0.00	1.64	0.03	24.02
		*Aristichthys nobilis*	P	UL	0.00	2.54	0.00	0.03	12.28	0.03
		*Hypophthalmichthys molitrix*	P	UL	0.00	1.02	0.00	3.04	0.04	0.00
Siluriformes	Siluridae	*Silurus asotus*	C	DM	0.00	1.52	6.25	0.00	0.00	0.00
	Bagridae	*Pelteobagrus fulvidraco **	C	DM	0.16	1.27	3.75	6.48	4.49	<0.01
Beloniformes	Hemiramphidae	*Hyporhamphus intermedius **	P	UL	0.00	0.00	0.00	<0.01	4.41	0.05
Perciformes	Gobiidae	*Tridentiger bifasciatus **	C	DM	28.76	11.17	11.25	32.19	28.25	0.03
		*Rhinogobius giurinus **	C	DM	2.26	0.00	2.50	4.07	2.52	47.57
		*Rhinogobius cliffordpopei **	C	DM	0.00	0.00	0.00	0.05	<0.01	23.38
	Channidae	*Channa argus*	C	LL	0.00	0.00	2.50	0.02	<0.01	3.07

C: carnivorous, O: omnivorous, P: planktivorous, H: herbivorous; UL: Upper-layer fish, LL: Lower-layer fish, DM: Demersal fish; DT: Datun Reservoir, DH: Donghu Reservoir, SWC: Shuangwangcheng Reservoir; *: small-body-size fish.

**Table 2 animals-13-01614-t002:** Relative importance index (IRI), relative abundance (N) and weight (W) percentages, and occurrence (F) percentage of dominant species detected by traditional fish landing (TFL) in three terminal reservoirs of the eastern route of the South-to-North Water Transfer Project.

Sampling Reservoirs	Dominant Species	IRI (×10^3^)	N (%)	W (%)	F (%)
DT	*Pseudobrama simoni*	5.71	13.09	44.05	100.00
	*Tridentiger bifasciatus*	3.61	28.76	7.33	100.00
	*Hemiculter leucisculus*	2.45	20.84	3.70	100.00
	*Abbottina rivularis*	2.08	16.48	4.30	100.00
	*Hemibarbus maculatus*	1.35	0.65	19.66	66.67
	*Pseudorasbora parva*	1.12	9.53	1.68	100.00
DH	*Acheilognathus macropterus*	5.14	39.59	11.76	100.00
	*Cultrichthys erythropterus*	4.35	9.14	34.32	100.00
	*Hemiculter leucisculus*	3.65	16.75	19.74	100.00
	*Carassius auratus*	1.43	7.61	13.89	66.67
	*Tridentiger bifasciatus*	1.26	11.17	1.43	100.00
SWC	*Silurus asotus*	4.04	6.25	54.29	66.67
	*Abbottina rivularis*	3.85	53.75	4.05	66.67
	*Protosalanx hyalocranius*	1.16	11.25	0.34	100.00
	*Channa argus*	1.16	2.50	32.21	33.33

DT: Datun Reservoir, DH: Donghu Reservoir, SWC: Shuangwangcheng Reservoir.

**Table 3 animals-13-01614-t003:** Diversity indices of fish assemblages detected by traditional fish landing (TFL) and environmental DNA (eDNA) metabarcoding in three terminal reservoirs of the eastern route of the South-to-North Water Transfer Project.

Sampling Reservoirs	Shannon–Wiener Index		Pielou’s Index		Chao1 Index		ACE Index	
	TFL	eDNA	TFL	eDNA	TFL	eDNA	TFL	eDNA
DT	1.91	1.59	0.73	0.55	15.00	19.00	14.74	20.04
DH	1.94	2.12	0.74	0.74	14.00	18.00	14.00	18.00
SWC	1.63	1.23	0.68	0.44	11.25	17.00	12.51	17.53

DT: Datun Reservoir, DH: Donghu Reservoir, SWC: Shuangwangcheng Reservoir.

## Data Availability

Data presented in this study are contained within the article.

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
