# Peer review of "Investigations of Fish Assemblages Using Two Methods in Three Terminal Reservoirs of the East Route of South-to-North Water Transfer Project, China"

_animals, 2023, doi:10.3390/ani13101614_

Round 1

Reviewer 1 Report

The authors of the manuscript compared fish composition and tradition and environmental DNA methods between three water reservoirs to monitor water quality. The study is interesting, well designed and well written, and is suitable for publication in this journal after addressing some minor comments.  

Line 51, Line 86: “m3” should be “m3

Lines 67, 322-323, 381, 385, 392: Scientific names are commonly italicized, “Hypophthalmichthys molitrix” and “Aristichthys nobilis

Line 85 and 404: typo, “SNWDP” should be “SNWTP”

Line 218: There is an erroneous “Figure 1” at the beginning of the sentence.  

Tables 1-3: There are inconsistencies with significant figures or number of decimal places in the reported values.

Author Response

Reply to the comments of the Reviewer #1

  • Line 51, Line 86: “m3” should be “m3”.

Reply: We corrected the mistakes (lines: 54, 59). We sincerely thank the reviewer for careful reading.

  • Lines 67, 322-323, 381, 385, 392: Scientific names are commonly italicized, “Hypophthalmichthys molitrix” and “Aristichthys nobilis”.

Reply: The scientific name has been changed to the italicized (lines: 81, 322, 386-387, 391, 398). Thanks for your correction.

  • Line 85 and 404: typo, “SNWDP” should be “SNWTP”.

Reply: We revised it (lines: 59, 411).

  • Line 218: There is an erroneous “Figure 1” at the beginning of the sentence.

Reply: We corrected the mistake (lines: 220). Thanks for your reminder.

  • Tables 1-3: There are inconsistencies with significant figures or number of decimal places in the reported values.

Reply: We have standardized the number of significant figures and the number of decimal places in the values (lines: 227-231,243-246,302-305).

Reviewer 2 Report

The MS compared two methods of fish assemblage monitoring in three terminal reservoirs of the east route of the South-to-North Water Transfer project, aiming to find a more suitable method for the management purpose of the terminal reservoirs. The study design was explained clearly. The novelty, the value, and the overall interest to readers is high.

The results include detailed comparison of the two methods. However, as a researcher who never worked on fish assemblage monitoring methods, I found the level of the agreement between the two methods not very high. I would suggest the authors to add some references on similar comparison studies of fish assemblage methods, preferably on the comparison between TFL and eDNA methods. Moreover, it would be more straightforward if the authors can develop/combine quantitative metrics to compare the two methods. By doing so the readers can have a better understanding of how different methods may differ from each other, and what kind of agreement is good in the application.

The MS can be improved by going through a more rigorous English editing process.

Author Response

Reply to the comments of the Reviewer #2

  • I found the level of the agreement between the two methods not very high.

Reply: Yes, the concordance of species detected simultaneously of eDNA with TFL were low, varies between 21,7 to 45,5% in the three reservoirs. The low congruence in species composition between eDNA metabarcoding and TFL results may be related to deficiencies in the taxonomic database used for eDNA referencing. Following the comment, we have it clarified at lines22-24, 37-39.

  • I would suggest the authors to add some references on similar comparison studies of fish assemblage methods, preferably on the comparison between TFL and eDNA methods.

Reply: Following the suggestion, we added another three references on similar comparative studies of fish assemblage between TFL and eDNA methods (lines: 370-373). These references tested performance of eDNA metabarcoding in monitoring fish diversity and concluded that results of eDNA metabarcoding were essentially similar and moderate level of congruence to that of traditional survey methods. we have it clarified at lines 356-359.

  • Moreover, it would be more straightforward if the authors can develop/combine quantitative metrics to compare the two methods. By doing so the readers can have a better understanding of how different methods may differ from each other, and what kind of agreement is good in the application.

Reply: We agreed with the comment and will consider developing quantitative metrics to compare the two methods in next research. However, it is difficult to establish the quantitative metrics in this study. We added this comment in our revision (lines:363-365).

  • The MS can be improved by going through a more rigorous English editing process.

Reply: Thanks for your suggestion. The ms had been polished by the Wiley Editing Services before its submission. Following the comment, we invited a native English speaker to further polish it.

Reviewer 3 Report

Dear Editors,

Dear Authors,

The manuscript entitled: “A Suitable Fish Monitoring Method Closely Depends on the Management Purposes of Terminal Reservoirs in the East Route of South-to-North Water Transfer Project, China” represents valuable study. The research aims to determine ichthyofauna composition in three terminal reservoirs of the East Route of the South-to-North Water Transfer Project, China by traditional fish landing (TFL) and environmental DNA (eDNA) metabarcoding methods. The obtained data provided baseline for future biomanipulation efforts aimed to conserve or improve water quality in the examined reservoirs. Moreover, the study verified the congruence between TFL and eDNA methods and checked their applicability for ichthyofauna monitoring in open waters. Despite, correct methodology the manuscript bears numerous flaws and major improvements are required. In my opinion the title does not reflect the study scope and aims should be revised. The results interpretation and conclusions in many places are not supported by the results – the main weakness is the notion that the results obtained by TFL and eDNA methods display similarity. All remarks about this have been placed in the attached file. Moreover, many terms, statements and imprecise sentences need to be explained better and revised. Language presentation require moderate corrections.

In conclusion, I do not recommend the manuscript for the publication in the present form and major revision should be made by authors. After that the manuscript should be evaluated again. All remarks, questions and fixes were placed in the attached pdf file (yellow highlights contain fixes and sentence suggestions, while red highlights contain comments and questions).

Thank you for another interesting manuscript that I could review!

Dear Editors,

Dear Authors,

The manuscript entitled: “A Suitable Fish Monitoring Method Closely Depends on the Management Purposes of Terminal Reservoirs in the East Route of South-to-North Water Transfer Project, China” represents valuable study. The research aims to determine ichthyofauna composition in three terminal reservoirs of the East Route of the South-to-North Water Transfer Project, China by traditional fish landing (TFL) and environmental DNA (eDNA) metabarcoding methods. The obtained data provided baseline for future biomanipulation efforts aimed to conserve or improve water quality in the examined reservoirs. Moreover, the study verified the congruence between TFL and eDNA methods and checked their applicability for ichthyofauna monitoring in open waters. Despite, correct methodology the manuscript bears numerous flaws and major improvements are required. In my opinion the title does not reflect the study scope and aims should be revised. The results interpretation and conclusions in many places are not supported by the results – the main weakness is the notion that the results obtained by TFL and eDNA methods display similarity. All remarks about this have been placed in the attached file. Moreover, many terms, statements and imprecise sentences need to be explained better and revised. Language presentation require moderate corrections.

In conclusion, I do not recommend the manuscript for the publication in the present form and major revision should be made by authors. After that the manuscript should be evaluated again. All remarks, questions and fixes were placed in the attached pdf file (yellow highlights contain fixes and sentence suggestions, while red highlights contain comments and questions).

Thank you for another interesting manuscript that I could review!

Author Response

Dear Reviewer:
We revised the ms according to your comments offered by the reviewer. A list of major revisions is described in attachment text. We are happy to make further revisions upon your request.
Thank.

Sincerely,

Wenping He, Fei Cheng

*********************************

Round 2

Reviewer 3 Report

Dear Editors,

Dear Authors,

The manuscript titled “Investigations of Fish Assemblages Using Two Methods in Three Terminal Reservoirs of the East Route of South-to-North Water Transfer Project, China” has undergone significant improvements in the Introduction, Materials and Methods, and Results chapters, requiring only minor revisions. However, the Discussion chapter would benefit from further revisions. In its current state, the chapter contains redundant information that may be confusing to readers and would benefit from additional clarification. Please refer to the attached file for all comments, questions, and suggested fixes, with yellow highlights indicating direct sentence suggestions and fixes, while red highlights signify comments.

Best regards,

Dear Editors,

Dear Authors,

The manuscript titled “Investigations of Fish Assemblages Using Two Methods in Three Terminal Reservoirs of the East Route of South-to-North Water Transfer Project, China” has undergone significant improvements in the Introduction, Materials and Methods, and Results chapters, requiring only minor revisions. However, the Discussion chapter would benefit from further revisions. In its current state, the chapter contains redundant information that may be confusing to readers and would benefit from additional clarification. Please refer to the attached file for all comments, questions, and suggested fixes, with yellow highlights indicating direct sentence suggestions and fixes, while red highlights signify comments.

Best regards,

Author Response

Dear editor and reviewer,
We are writing to submit the revision of our manuscript, animals-2356611, Investigations of fish assemblages using two methods in three terminal reservoirs of the East Route of South-to-North Water Transfer Project, China. Our manuscript was revised according to the comments of the reviewer. A list of revisions is described below.

We are happy to make further revisions upon your request.
Thank you very much for your attention.

Sincerely,

Wenping He, Fei Cheng
